# Experience-Evolving Multi-Turn Tool-Use Agent with Hybrid Episodic-Procedural Memory

**Sijia Li** [1 2 3]  **Yuchen Huang** [1]  **Zifan Liu** [1]  **Zijian Li** [1]
**Jingjing Fu** [2]  **Lei Song** [2]  **Jiang Bian** [2]  **Jun Zhang** [1]  **Rui Wang** [2]

## Abstract

As intents unfold and environments change, multiturn agents face continuously shifting decision contexts. Although reusing past experience is intuitively appealing, existing approaches remain limited: full trajectories are often too context-specific to transfer, while tool-level reuse ignores the context and environment. In this paper, we introduce a hybrid episodic–procedural memory strategy (H-EPM) that enables experience-evolution of multi-turn tool-use policies, by adaptively reusing partially overlapping successful experiences in both inference and training. Inspired by human episodic–procedural integration, we build a tool graph from accumulated trajectories, where recurring tool-to-tool dependencies capture procedural routines and each edge is augmented with a compact episodic summaries of relevant context. At inference, the agent dynamically balances episodic recall for contextual reasoning and procedural execution for routine steps. Beyond inference, H-EPM introduces a memory-guided reinforcement learning paradigm that directly addresses a core challenge in multi-turn agent RL: ineffective exploration over long trajectories. By biasing exploration toward historically successful tool transitions, H-EPM learns a stronger policy that generalizes during inference without relying on domain-specific experience collection. Experiments show that H-EPM consistently delivers substantial inference-time gains over strong baselines across multi-turn tool-use benchmarks, reaching up to 50%+. It also boosts RL policy performance, achieving up to 40%+ improvement on out-of-distribution tasks. Our code is available

[1]Hong Kong University of Science and Technology [2]Microsoft Research Asia [3]Work done during internship at Microsoft. Correspondence to: Jun Zhang <eejzhang@ust.hk>, Rui Wang <wrui0920@gmail.com>.

*Proceedings of the 43rd International Conference on Machine Learning*, Seoul, South Korea. PMLR 306, 2026. Copyright 2026 by the author(s).

at `https://github.com/LISijia-dev/H-EPM`.

## 1. Instruction

Large language models (LLMs) demonstrate strong reasoning and decision-making capabilities. However, many real-world tasks cannot be addressed by reasoning alone but require external tools, such as querying databases for up-to-date facts or invoking a code interpreter for computation (Patil & Jadon, 2025; Huang et al.). Accordingly, typical agent architectures augment an LLM with planning, memory, and tool-use modules, with the LLM as a central controller. While such systems perform well in single-turn settings with fully specified queries (Yuan et al., 2024), real applications are inherently multi-turn, where user intent is iteratively clarified, and intermediate tool calls change the environment. Decision quality therefore depends not only on the current prompt but also on the dialogue history and the sequence of prior tool invocations (Guan et al., 2025). The multi-turn scenario introduces partial observability, making early tool selection error-prone and requiring agents handle changing state induced by dialogue history and prior tool invocations, while adapting plans as new evidence arrives.

When humans tackle complex tasks, they recognize recurring situations and selectively reuse elements of past successes to guide future decisions. In neurobiology, these forms of reuse are often distinguished as episodic memory and procedural memory (Atallah et al., 2004). *Episodic memory*, supported by hippocampal–midbrain interactions, integrates overlapping episodes into linked mnemonic structures that extend beyond single events and enable flexible generalization (Shohamy & Wagner, 2008). *Procedural memory* is supported by the basal ganglia and allows people to acquire skills through repeated practice and reinforcement of prior actions (Wise, 1996). Recent experience-based LLM agents echo these two forms of memory in different ways. Methods that store and retrieve prior task-solving trajectories (Ouyang et al., 2025; Zhang et al., 2024) behave episodically. Given a new query, they recall whole trajectories or predefined subtasks. Other works construct

tool graphs to repeat successful tool-use patterns (Liu et al., 2024a), emphasizing procedural regularities. However, both coarse-grained episodes and tool-only dependencies are ill-suited to multi-turn scenarios. Coarse episodes treat each trajectory or predefined subtask as an indivisible unit, preventing reuse and adaptation under partially overlapping experiences, while tool-only links ignore how dialogue history and earlier tool calls shape the next action. Inspired by human decision-making arising from the interaction between episodic and procedural memory rather than either in isolation, our framework couples episodic-like fragments with procedural-like routines: it retrieves partially overlapping trajectory segments while abstracting stable tool-use dependencies from repeated sequences of tool invocations, yielding context-consistent reuse of past experience in continually evolving multi-turn tool-use scenarios.

In this paper, we propose a hybrid Episodic–Procedural memory strategy, **H-EPM**, that enables multi-turn tool-use agents to evolve through experience by adaptively leveraging past trajectories in partially overlapping situations during both inference and RL rollouts. H-EPM first induce a tool graph from accumulated tool-use trajectories, where edges capture recurring tool invocation patterns. However, tool-level transitions alone are insufficient for multi-turn decision-making, as they ignore task-specific information. To address this, we augment edges with compact state descriptors that summarize decision-relevant dialogue and tool history, enabling retrieval and adaptation across partially overlapping situations. At inference, H-EPM enables adaptive retrieval between episodic and procedural memory. When contextual recall is required, the agent retrieves state summaries from relevant edges to guide tool selection; otherwise, it relies on high-confidence transition weights for routine decisions. By jointly leveraging transferable tool dependencies and state-specific guidance from similar past experiences, the agent achieves more effective tool selection in open-ended multi-turn interactions. Beyond inference, H-EPM further extends to a memory-guided Reinforcement Learning (RL) paradigm in which exploration is progressively improved by an evolving memory, allowing historical trajectories to be adaptively reused under partially overlapping conditions and driving sustained policy improvement. We summarize the contributions of our work as follows:

- We propose H-EPM, a hybrid Episodic–Procedural Memory framework that enables multi-turn agents to evolve from partially overlapping experiences. H-EPM consolidates successful trajectories into a structured memory graph, where nodes correspond to tools and edges encode both procedural transition patterns and compact episodic state summaries. At inference time, the agent jointly exploits these two forms of memory to guide tool selection, enabling experience-driven decision making on new tasks.

- We integrate H-EPM into RL training as a memory-guided exploration mechanism to tackle a key challenge in multi-turn agent RL: long trajectories often result in ineffective exploration. H-EPM guides rollouts toward high-quality trajectories by prioritizing actions that historically led to success, enabling the agent to learn more effective policies. The resulting policy generalizes to unseen tasks and tools at inference without requiring experience collection.

- Experiments show that H-EPM consistently delivers substantial inference-time gains of up to 50%+ over strong baselines across multi-turn tool-use benchmarks. Additionally, integrating H-EPM into reinforcement learning enhances policy optimization, yielding up to 40%+ improvement on out-of-distribution tasks.

## 2. Related work

### 2.1. Multi-turn Tool Use

Multi-turn tool use is challenges as agents should not only select the appropriate tool but also manage the intricate interaction in dialogue and tool outputs across successive turns. To enable such capabilities, prior work has explored both supervised fine-tuning (SFT) and reinforcement learning (RL) approaches. SFT methods generate synthetic multi-turn data via graph translation (Yin et al., 2025) and simulated interactions (Prabhakar et al., 2025). RL methods focus on optimizing tool-use policies through interactive feedback. For example, MUA-RL (Zhao et al., 2025) and RAGEN (Wang et al., 2025b) study multi-turn tool use with user interactions and self-evolution in text-based environments, respectively, while verl-agent (Feng et al., 2025) and Tool-Star (Dong et al., 2025) extend the RL post-training paradigm to more complex reasoning and multi-tool settings. Complementary to these algorithmic advances, recent benchmarks such as DialogTool (Wang et al., 2025a) and ToolSandbox (Lu et al., 2024) highlight the difficulties of maintaining state across turns and call for systematic evaluation of tool-use agents. Nevertheless, existing SFT and RL remain limited in how past multi-turn tool-use experience is represented and reused. SFT relies on fixed, pre-collected trajectories, while RL often explores long-horizon tool sequences with sparse guidance, making learning ineffective and struggling to adapt when the dialogue context changes.

### 2.2. Agent Memory

Agent memory has been explored in two primary ways: internal and external. Internal memory is generated and maintained within the agent's reasoning or planning modules, often serving as intermediate representations to guide decision-making (Lee et al., 2024; Agashe et al., 2024; Jiang et al., 2024; Song et al., 2024). In contrast, external memory stores distilled information outside the agent's core

reasoning loop, enabling modular retrieval and reuse across different tasks and time steps (Zhang et al., 2025b; Xu et al., 2025; Rezazadeh et al., 2024; Wang et al., 2023; Tang et al., 2025). Recent studies increasingly emphasize this external approach for its flexibility and scalability in long-horizon interactions, as it can retain a larger amount of past experience to support continual learning and adaptive behavior.

Besides, memory systems exhibit diverse forms, including pure experiential records (Xiong et al., 2025), summarized textual content (Ouyang et al., 2025), latent embedding representations (Zhang et al., 2025a), and structured graph-based memory architectures (Chhikara et al., 2025; Liu et al., 2025; Anokhin et al., 2024) (More details can be found in Appendix A.). Prior approaches mainly operate at the trajectory level, limiting their flexibility in multi-turn tool-use settings. In contrast, we model decision-making at the state level, enabling experience transfer across similar states. Our approach leverages a hybrid episodic–procedural memory to adaptively retrieve relevant prior experiences according to both procedural and state similarity, thereby providing efficient and context-aware memory-based guidance.

## 3. Memory-Augmented Task Formulation

We formulate multi-turn tool use as a partially observable Markov decision process (POMDP) $(\mathcal{E}, \mathcal{O}, \mathcal{A}, T, \pi_\theta)$, where $\mathcal{E}$ is the latent dialogue–environment state, $\mathcal{A}$ comprises natural language responses and tool invocations, and $\mathcal{O}$ consists of dialogue utterances and tool feedback. Environment dynamics are governed by $T(e_{t+1} \mid e_t, a_{t+1})$, while a language-model policy $\pi_\theta(a_{t+1} \mid h_t)$ acts on the observable interaction history $h_t$, reflecting partial observability of $e_t$. At step $(t+1)$, the agent samples an action $a_{t+1} \sim \pi_\theta(\cdot \mid h_t)$, observes $o_{t+1}$, updates its history $h_{t+1} = (h_t, a_{t+1}, o_{t+1})$. Tool invocations may alter the latent state, with effects revealed only indirectly through future observations.

**Episodic-Procedural Memory** Storing full observations or interaction histories leads to overly specific experiences that generalize poorly. To enable effective reuse, we construct experience-based memories over *abstracted tool-level transitions*, rather than raw observations.

- Episodic memory stores context-conditioned transitions $(s_t, a_t) \Rightarrow a_{t+1}$, where $s_t = \phi(h_t)$ is a compact belief summary capturing effective next-tool choices under similar partial observations.

- Procedural memory stores context-free transitions $a_t \Rightarrow a_{t+1}$, encoding recurrent tool-to-tool execution patterns that generalize across tasks.

## 4. Methodology

In this section, we first explain how the historical experience is obtained (Section 4.1). We then use this experience to construct the memory in H-EPM, which integrates weighted and summary information on the edges to store both the procedural memory and episodic memory (Section 4.2) in the graph structure. Finally, we explain how to adaptively guide the agent's tool-selection decisions in both inference and RL training for agent system (Section 4.3).

### 4.1. Historical Experience Collection

In multi-turn tool-use, task-relevant information is dispersed across long and heterogeneous interaction histories. To extract reusable experience, we compress each history into a summary-based state $s_t = \phi(h_t)$ that capture decision-relevant information while abstracting away low-level details. These summarized states form the basis of episodic memory, enabling reuse across similar partial observations.

To avoid per-turn summarization overhead, we introduce an autonomous **state-summarization tool** . Rather than enforcing fixed-step abstraction, the agent decides when summarization is required by invoking the tool. Each invocation yields a compact state information (as in Fig. 1) that supports both decision making and experience storage.

At summarization steps, the agent selects actions conditioned on the dialogue context and the summarized state: $a_{t+1} \sim \pi_\theta(a_{t+1} \mid h_t, s_t)$. The resulting tuples $(s_t, a_t, a_{t+1})$ form the core of episodic experience.

We generate tool-augmented trajectories by rolling out the base model on the training set with automatic summarization tool. These trajectories provide the historical experience used to construct episodic–procedural memory and are shared across inference-time optimization and RL training. When no training set is available, test-time trajectories can be incorporated online. During RL training, additional experience is collected throughout the rollout process.

### 4.2. Memory Construction for H-EPM

To capture both procedural tool-transition patterns and episodic state information, we organize the hybrid memory into a state-annotated tool-transition graph $G = (V, E, W, S)$, where each node $\tau \in V$ corresponds to a tool-type action $\tau \in \mathcal{A}_{\text{tool}} \subset \mathcal{A}$. The associated weight $w(\tau_t, \tau_{t+1}) \in W$ encodes procedural statistics such as transition frequency or empirical utility, while the episodic annotation $s_t \in S$ captures summarized states observed under similar contextual situations. As illustrated in Fig. 1, the graph is constructed from historical trajectories, including tool invocations and summarized states of successful tasks.

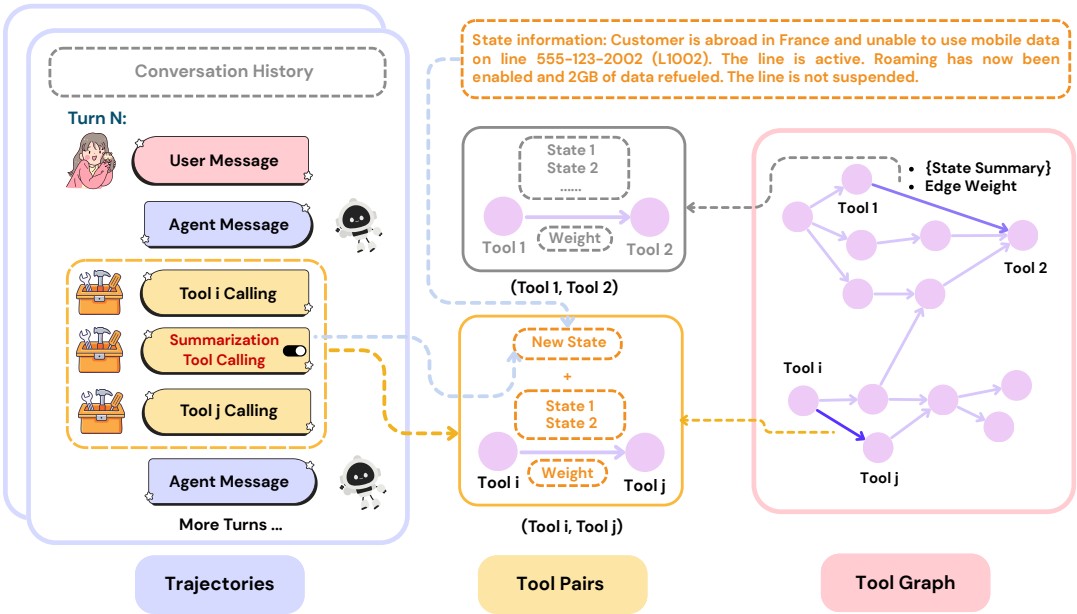

*Figure 1.* **Building the memory graph in H-EPM.** We build the graph based on previous successful trajectories. Except for the summarization tool, all tools invoked in historical trajectories are treated as nodes, and edges are constructed to connect them according to their calling sequential relationships. The state information summarized by the summarization tool is treated as edge information. Each edge must contain a weight and may additionally include the corresponding state information, depending on whether the agent chooses to invoke the summarization tool between the two tool nodes.

**Episodic Memory** stores state-action associations observed in similar contexts, capturing which tool choices succeeded under partial observations. Specifically, when a state-summarization tool is invoked between two tool actions, the resulting summarized state $s_t = \phi(h_t)$ is stored as an episodic annotation attached to the corresponding edge. Each edge can accumulate multiple summarized states observed between the same tool transitions. For example, consider a historical trajectory: $(..., m_{user}, a_{t-1} = \tau_1, \ a_t = \tau_{sum}, \ a_{t+1} = \tau_2, \ m_{agent}, ...)$, where $m_{user}$ and $m_{agent}$ denote messages from the user and the agent respectively, and $\tau_{sum}$ denotes the state-summarization tool. Invoking $\tau_{sum}$ produces a summarized state $s_t = \phi(h_t)$, which is then attached to the edge $(\tau_1, \tau_2)$ as episodic experience: $(\tau_1, \ \tau_2, \ w(\tau_1, \tau_2), \ s_t)$.

**Procedural Memory** captures statistical patterns of tool transitions, enabling the agent to learn common or effective action sequences. Each directed edge $(\tau_t, \tau_{t+1}) \in E$ represents a historical transition between two tool actions. The weight represents both the success rate of the corresponding tasks and the efficiency of successful trajectories. Previous work typically relies solely on the probability of successful transformation between tools to define edge weights (Liu et al., 2024a). However, this approach may lead to the tool overuse trap. To mitigate this issue, we incorporate efficiency as an additional weighting factor, enriching the information encoded in the graph. The efficiency is mea-

sured by the number of rollout steps required to complete the task. By leveraging the accuracy–efficiency tradeoff, the agent can develop resource-bounded strategies that avoid repeatedly invoking tools that are harmless but task-irrelevant, such as simple information retrieval tools.

To be specific, we define the weight as:

$$w'(\tau_t, \tau_{t+1}) = N(\tau_t, \tau_{t+1}) + c \sum_{k=1}^{N(\tau_t, \tau_{t+1})} \frac{1}{n_k(\tau_t, \tau_{t+1})},$$

where $N(\tau_t, \tau_{t+1})$ denotes the number of successful trajectories in which the transition from tool $\tau_t$ to $\tau_{t+1}$ occurs, $n_k(\tau_t, \tau_{t+1})$ is the total number of rollout steps of the $k$-th such successful trajectory, and $c$ controls the relative importance of efficiency.

To ensure local comparability across candidate next tools, the outgoing edge weights are normalized for each tool: $w(\tau_t, \tau_{t+1}) = \frac{w'(\tau_t, \tau_{t+1})}{\sum_{\tau' \in \mathcal{N}(\tau_t)} w'(\tau_t, \tau')}$, where $\mathcal{N}(\tau_t)$ denotes the set of admissible next-tool actions following $\tau_t$. This normalization ensures that outgoing edge weights sum to one, representing the relative importance of tool dependencies conditioned on the current tool.

### 4.3. Experience-Driven Evolution with H-EPM

**H-EPM Enhanced Inference.** H-EPM can provide tool-selection guidance for the agent, as shown in Fig.2. After

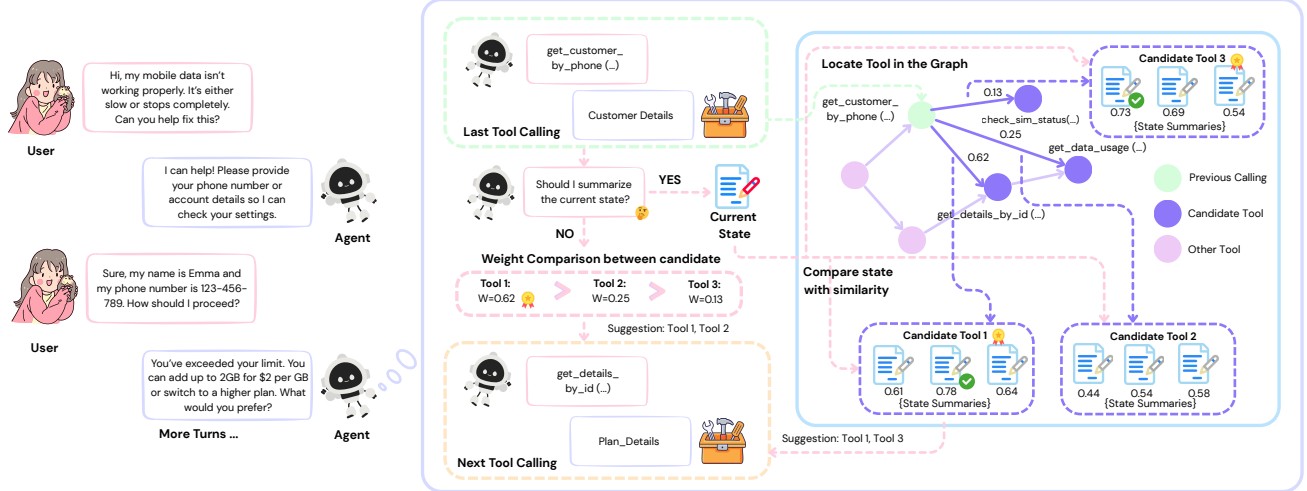

*Figure 2.* **Adaptive experience retrieval with H-EPM.** First, we locate the last tool call in the graph and identify the candidate node it is connected to. Then, the agent decides whether it should summarize the current state. If the decision is **yes**, the agent calls the summary tool and abstracts the current state as a summary. It then compares the similarity between this summarized state and the stored states in the connected edges, selecting the two with the highest similarities as the next tool candidates. If the decision is **no**, the agent directly selects the next two tools by comparing the edge weights. Both the similarity comparison and the weight comparison methods return two candidate tools for suggestion.

each tool invocation, the corresponding tool node is located in the graph, and its adjacent tools are considered candidates for the next step. Before acting, the agent first decides whether to invoke the state-summarization tool. If invoked, the resulting state summary is compared against the summaries stored on edges connecting to adjacent tools. The top-k tools with the highest similarity scores are suggested, effectively leveraging episodic memory to recall context-relevant past experiences. Otherwise, the agent relies solely on edge weights and selects the top-k adjacent tools with the highest transition weights, which corresponds to exploiting procedural memory capturing frequent tool-transition patterns. Providing top-k candidates rather than a single choice preserves flexibility and generality in decision-making. A more detailed case study is provided in Appendix B.

In summary, H-EPM adaptively switches between episodic and procedural memory: it exploits high-level context through episodic memory when state summarization is invoked, while relying on procedural memory to follow frequent tool-transition patterns. This hybrid-memory design enables flexible and context-aware tool selection, allowing the agent to effectively handle diverse and evolving task scenarios, exhibiting strong generalization ability when encountering new tasks.

**H-EPM Enhanced RL.** We integrate H-EPM into the rollout process to guide exploration in multi-turn agent RL, addressing the challenge that long trajectories often hinder effective exploration. H-EPM enables the agent to prioritize actions that have historically led to success, effectively

increasing the probability of discovering a successful trajectory without exhaustive random exploration.

Unlike inference, RL requires active exploration, and over-reliance on past experience can limit discovery of new strategies. While LLMs implicitly encode episodic and procedural knowledge, long-horizon multi-turn tasks can make episodic signals less apparent, making it harder to maintain coherent trajectories. H-EPM mitigates this challenge by providing episodic guidance during training, while modulating the influence of procedural tool-to-tool guidance via an adjustable skip rate. The overall decision process mirrors the stepwise tool selection used at inference time. When procedural memory is activated, the top-$k$ suggested tools are randomly skipped with probability $p_{\text{skip}}$, injecting stochasticity that promotes effective exploration.

This mechanism allows the agent to utilize past experience while exploring new actions. The memory graph is updated online with newly successful trajectories, allowing the guidance mechanism to evolve alongside training. By incorporating memory during training, the resulting policy generalizes to unseen task domains, reducing the need for task-specific experience collection at inference.

## 5. Experiments

### 5.1. Experimental Settings

**Benchmarks.** To verify the effectiveness of our method, we conduct evaluations on four multi-turn tool-use benchmarks to demonstrate the superiority of our approach. We

select Telecom tasks in $\tau$-Bench (Yao et al., 2024), Retail tasks in $\tau^2$-Bench (Barres et al., 2025), ToolSandbox (Lu et al., 2024) and agent multi-turn task in ACEBench (Chen et al., 2025a), which offer diverse stateful, conversational, and interactive real-world scenarios, in which an LLM-simulated user interacts with the LLM agent. Among these benchmarks, $\tau$-Bench (Telecom) and $\tau^2$-Bench (Retail) provide separate training and test sets, with the training data used to construct the tool graph. ToolSandbox includes only a test set of approximately 1,000 tasks. We therefore split the data into approximately 800 training instances and 200 test instances, using the training set to construct the memory. When the training set is unavailable, we update memory using only inference-time data during testing and evaluate on ACEBench. Further details are provided in Appendix C.1.

**Evaluation.** To ensure a fair comparison, we follow the evaluation protocols defined in the original benchmark settings. Specifically, $\tau^2$-Bench uses assertion functions to evaluate task success, $\tau$-Bench evaluates performance by comparing the final database state of each episode with the ground-truth expected state, and ToolSandbox adopts a milestone-based evaluation strategy that defines key events which must occur within a trajectory. Despite differing evaluation criteria, we use test-set accuracy as a unified performance metric across benchmarks.

**Implementation Details** When adapting H-EPM, we employ two different strategies that allow the agent to decide when to summarize the current state, as detailed in Appendix C.4. For pure inference, we primarily adopt an explicit strategy, whereas for RL training, we use an implicit strategy. For RL training, we adopt Group Relative Policy Optimization (GRPO) (Shao et al., 2024) as the base algorithm and integrate H-EPM into the rollout process to guide exploration. Except for the rollout stage, all other settings remain identical to the standard GRPO. Training is conducted in a multi-turn, user-interactive framework based on VolcEngine Reinforcement Learning (VeRL) (Sheng et al., 2025). For training data, we use 500 training tasks from the $\tau$-bench Retail benchmark. Qwen3-4B-Instruct-2507 is adopted as the primary backbone model in our experiments. More details are provided in Appendix E.

**Baselines.** We compare our method with strong baselines for both inference and RL. **For inference,** we compare our method with a basic retrieval approach, where the top three past experiences most similar to the current task are retrieved. In addition, we include comparisons with the memory-based method ReasoningBank (Ouyang et al., 2025), where the agent retrieves summaries of previous similar trajectories, and the tool graph–based method ToolNet

(Liu et al., 2024a) [1], which organizes tools into a directed graph, allowing the agent to navigate it by iteratively selecting the next tool from its successor nodes until the task is fully resolved. **For RL,** we consider the standard GRPO algorithm and an SFT model trained on trajectories generated by H-EPM, and we further compare our method against AgentEvolver (Zhai et al., 2025).

## 5.2. Main Results

**H-EPM Enhanced Inference.** As shown in Table 1, H-EPM achieves consistent and significant performance gains across benchmarks with different base models when compared with both episodic memory and tool graph based method. Notably, the performance improvements are more pronounced for base models with weaker reasoning capabilities. GPT-4.1-mini benefits more from our method than GPT-4.1 and GPT-4o in most settings. Moreover, Qwen3-4B-Instruction also exhibits stronger instruction-following capability, whereas Qwen3-8B demonstrates superior reasoning capacity. This suggests that our approach is particularly effective at compensating for the reasoning limitations of smaller or less capable models with adaptive memory. While $\tau$-Bench and $\tau^2$-Bench focus on single-domain tasks, ToolSandbox spans multiple domains. Our method consistently outperforms all baselines across these benchmarks, demonstrating strong generalizability to varying task complexity and tool diversity. Among the baselines, step-level methods (e.g., ToolNet) outperform trajectory-level retrieval (e.g., Reasoning Bank), reflecting the incremental revelation of relevant information in multi-turn interactions. Trajectory-level retrieval may introduce incomplete or even inaccurate contextual signals during early reasoning stages. By adaptively combining episodic and procedural experience, our method delivers more reliable guidance and superior performance.

Recent powerful models achieve notable gains on $\tau^2$-Bench. To explore the effectiveness of our approach under high reasoning capabilities, we conducted experiments on GPT-5.1 high-reasoning, and the results are demonstrated in Table 2. Although GPT-5.1 already demonstrates strong performance on benchmark tasks, integrating H-EPM further guides the agent toward more effective decision-making, resulting in consistently higher overall performance.

**H-EPM Enhanced RL.** As shown in Table 3, equipping GRPO with H-EPM yields consistently larger performance gains and stronger cross-benchmark generalization than the base GRPO and AgentEvolver. By contrast, SFT on successful H-EPM trajectories may overfit and hurt performance. While AgentEvolver also exploits prior experience, it oper-

---

[1] Since the original implementation is not open-sourced, we re-implemented it ourselves.

*Table 1.* Performance comparison of different models across three benchmarks, showing the average task reward (1 for completed tasks, 0 for incomplete tasks) and relative improvement over the Base Model. Qwen3-4B-Ins: Qwen3-4B-Instruct-2507.

| BENCHMARK | MODEL | BASE MODEL | REASONING BANK | TOOLNET | RETRIEVAL TOP-3 | H-EPM |
|---|---|---|---|---|---|---|
| $\tau$-BENCH | GPT-4.1-MINI | 0.541 | 0.567 (+4.8%) | 0.572 (+5.7%) | 0.575 (+6.3%) | **0.661 (+22.2%)** |
| | GPT-4.1 | 0.658 | 0.645 (-2.0%) | 0.652 (-0.9%) | 0.672 (+2.1%) | **0.710 (+7.9%)** |
| | GPT-4O | 0.609 | 0.624 (+2.5%) | 0.634 (+4.1%) | 0.644 (+5.7%) | **0.670 (+10.0%)** |
| | QWEN3-4B-INS | 0.435 | 0.451 (+3.7%) | 0.458 (+5.3%) | 0.451 (+3.7%) | **0.496 (+14.0%)** |
| | QWEN3-8B | 0.383 | 0.376 (-1.8%) | 0.398 (+3.9%) | 0.344 (-10.2%) | **0.421 (+9.9%)** |
| $\tau^2$-BENCH | GPT-4.1-MINI | 0.396 | 0.441 (+11.4%) | 0.452 (+14.1%) | 0.422 (+6.6%) | **0.526 (+32.8%)** |
| | GPT-4.1 | 0.358 | 0.413 (+15.4%) | 0.425 (+18.7%) | 0.439 (+22.6%) | **0.512 (+43.0%)** |
| | GPT-4O | 0.287 | 0.309 (+7.7%) | 0.333 (+16.0%) | 0.298 (+3.8%) | **0.362 (+26.1%)** |
| | QWEN3-4B-INS | 0.158 | 0.184 (+16.5%) | 0.193 (+22.2%) | 0.174 (+10.1%) | **0.232 (+46.8%)** |
| | QWEN3-8B | 0.201 | 0.246 (+22.4%) | 0.256 (+27.4%) | 0.189 (-6.0%) | **0.309 (+53.7%)** |
| TOOLSANDBOX | GPT-4.1-MINI | 0.584 | 0.628 (+7.5%) | 0.624 (+6.8%) | 0.582 (-0.3%) | **0.658 (+12.7%)** |
| | GPT-4.1 | 0.633 | 0.605 (-4.4%) | 0.643 (+1.6%) | 0.526 (-16.9%) | **0.704 (+11.2%)** |
| | GPT-4O | 0.619 | 0.632 (+2.1%) | 0.652 (+5.3%) | 0.598 (-3.4%) | **0.673 (+8.7%)** |
| | QWEN3-4B-INS | 0.478 | 0.532 (+11.3%) | 0.549 (+14.9%) | 0.542 (+13.4%) | **0.565 (+18.2%)** |
| | QWEN3-8B | 0.489 | 0.476 (-2.7%) | 0.498 (+1.8%) | 0.482 (-1.4%) | **0.514 (+5.1%)** |

*Table 2.* Performance comparison of different models across five benchmarks.

| Model | $\tau^2$-Bench | $\tau$-Bench | ToolSandbox |
|---|---|---|---|
| GPT-5.1 base | 0.842 | 0.739 | 0.647 |
| GPT-5.1 with H-EPM | 0.921 | 0.791 | 0.670 |

ates at the trajectory-level, which degrades as trajectories become longer and less transferable in multi-turn settings. In contrast, incorporating episodic–procedural memory during RL facilitates fine-grained reuse of experience fragments, allowing the agent to leverage partially overlapping past interactions to guide exploration.

The effectiveness of H-EPM further depends on the skip rate $p_{skip}$, which controls the extent to which procedural tool-transition suggestions are utilized. In ToolSandbox, tools exhibit strong sequential dependencies: for instance, open the location service is almost always followed by get the current location. Such domain benefits most from incorporating richer procedural experience, with optimal performance achieved at $p_{skip} = 0.8$. In contrast, $\tau$-Bench features weaker tool-to-tool coupling: after retrieving order information, the agent may branch into multiple valid actions (e.g., initiating a return or modifying order details). Accordingly, a higher skip rate ($p_{skip} = 0.9$) yields the best results. Finally, tasks in $\tau^2$-Bench primarily involve state-dependent error checking. In this case, entirely skipping weight-based tool suggestions ($p_{skip} = 1.0$) is most effective.

Overall, H-EPM consistently outperforms all baselines across skip rates ranging from 0.8 to 1.0. Moreover, adjusting $p_{skip}$ to match the degree of tool-to-tool reliance in a given domain enables H-EPM to achieve stronger task-specific performance. In addition, H-EPM only incurs negligible RL training overhead (Detailed in Appendix D).

### 5.3. Ablation Study

In this section, we conduct ablation studies to analyze the key design choices of H-EPM. Since H-EPM enhanced inference and RL share the same memory mechanism, we perform most ablations at the inference stage. For reinforcement learning, we focus specifically on ablations of the rollout strategy.

**The Components of H-EPM** We conduct ablation studies to assess the contribution of each component in the H-EPM framework. We consider three variants: (i) w/o memory: the base model is equipped only with the summarization tool, which the agent can invoke autonomously, without any memory based guidance. We also analysis the mechanism of the summarization tool with entropy in Appendix C.3. (ii) w/o wight: the memory guidance is conditioned solely on the current state representation, ignoring edge weights and tool-dependency signals. This setting evaluates whether state-conditioned retrieval (episodic memory) alone can provide sufficient guidance. (iii) w/o state: the memory guidance only follows the edge weights, ranking tools accordingly and retrieving the top-$k$ candidates based solely on edge weights (procedural memory). The results reported in Table 4(A) show that, even without the graph structure, the agent's adaptive use of the summarization tool consistently yields performance gains across different models and benchmarks. This variant isolates the contribution of memory summarization from that of graph-based retrieval. Moreover, the results indicate that both state information and edge-weight information within the H-EPM memory contribute significantly to the final performance.

**The Design of Weight** To assess the effectiveness of our weight design, we conduct two experiments: (i) Only weight. In this setting, we provide suggestions based solely on the top-$k$ weight values with both accuracy and efficiency in-

*Table 3.* Performance comparison across benchmarks for RL training results with Qwen3-4B-Instruct-2507. AE: AgentEvolver.

| BENCHMARK | BASE | GRPO | SFT | AE | H-EPM($p_{skip}$=0.8) | H-EPM($p_{skip}$=0.9) | H-EPM($p_{skip}$=1.0) |
|---|---|---|---|---|---|---|---|
| $\tau$-BENCH | 0.435 | 0.510 (+17.2%) | 0.373 (-14.3%) | 0.520 (+19.5%) | 0.534 (+22.8%) | **0.558 (+28.3%)** | 0.544 (+25.1%) |
| $\tau^2$-BENCH | 0.158 | 0.178 (+12.7%) | 0.163 (+3.2%) | 0.167 (+5.7%) | 0.218 (+38.0%) | 0.213 (+34.8%) | **0.223 (+41.1%)** |
| TOOLSANDBOX | 0.478 | 0.503 (+5.2%) | 0.473 (-1.0%) | 0.490 (+2.5%) | **0.522 (+9.2%)** | 0.505 (+5.6%) | 0.510 (+6.7%) |

*Table 4.* Ablation results for H-EPM enhanced inference.

| METHOD | $\tau^2$-BENCH | $\tau$-BENCH | TOOLSANDBOX |
|---|---|---|---|
| BASE MODEL | 0.358 | 0.658 | 0.633 |
| H-EPM | 0.512 | 0.710 | 0.704 |
| **(A) COMPONENT ABLATIONS** | | | |
| (I) W/O MEMORY | 0.430 | 0.667 | 0.647 |
| (II) W/O WEIGHT | 0.472 | 0.684 | 0.663 |
| (III) W/O STATE | 0.465 | 0.678 | 0.674 |
| **(B) WEIGHT DESIGN ABLATIONS** | | | |
| (I) ONLY WEIGHT | 0.465 | 0.678 | 0.674 |
| (II) - EFFICIENCY | 0.456 | 0.660 | 0.655 |
| **(C) GRAPH-LEVERAGING METHODS** | | | |
| (I) W/O ADAPT | 0.477 | 0.652 | 0.678 |
| (II) WITH GNN4PLAN | 0.427 | 0.669 | 0.602 |
| **(D) TOP-K TOOL SUGGESTION** | | | |
| (I) K=1 | 0.481 | 0.669 | 0.684 |
| (II) K=2 | 0.512 | 0.710 | 0.704 |
| (II) K=3 | 0.501 | 0.687 | 0.689 |

*Table 5.* Adapting H-EPM with half trajectories.

| METHOD | $\tau^2$-BENCH | $\tau$-BENCH | TOOLSANDBOX |
|---|---|---|---|
| GRPO | 0.178 | 0.510 | 0.503 |
| HALF | 0.202 | 0.528 | 0.496 |
| ALL | 0.223 | 0.544 | 0.510 |

Selective retrieval of episodic and procedural memory is crucial for effective multi-turn decision making. The performance degradation observed in ablated variants demonstrates that both memory adaptation and fine-grained state awareness jointly contribute to the robustness and efficiency of H-EPM.

**The Selection of $k$ in the Top-$k$ Tool Suggestion** As shown in Table 4(D), we evaluate values of $k \in \{1, 2, 3\}$, corresponding to returning the Top-$k$ tools as suggestions when the retrieval criteria are met. The results show that performance degrades when only one or three tool candidates are suggested. Selecting a single tool is overly restrictive, while providing three candidates introduces additional noise. Consequently, top-2 selection achieves the best trade-off and yields the optimal performance.

**Rollout Strategy** We evaluate a mixed-group setting in which only half of the rollouts are equipped with H-EPM, as in AgentEvolver (Zhai et al., 2025). This setting primarily studies the impact of episodic memory ($p_{\text{skip}} = 1$), since its effect is analogous to the skip rate in encouraging unguided actions under procedural memory. The results in Table 5 indicate that applying H-EPM to all rollouts yields superior performance. When H-EPM is used for all rollouts, a large fraction of actions are still generated without guidance, as suggestions are invoked only on demand by the agent, thereby preserving sufficient exploration.

## 6. Conclusions

In this paper, we propose H-EPM for multi-turn agents that leverages decomposed prior experience to support both inference and RL learning. Across diverse and challenging benchmarks, H-EPM consistently outperforms strong baselines. When integrated into RL, H-EPM guides long-horizon exploration and internalizes effective decision strategies in the policy, yielding strong generalization at inference without requiring domain-specific experience collection. More generally, H-EPM addresses a fundamental challenge

formation. (ii) Accuracy-only weighting. In this variant, the memory guidance only depend on the accuracy signal adopted as in prior work (Liu et al., 2024a), excluding the efficiency term introduced by our method. This allows us to isolate the contribution of the efficiency component. Additionally, we study the effect of the weight parameter $c$ in Appendix C.2. As shown in Table 4(B), all ablated variants exhibit performance degradation compared to the full H-EPM strategy. The results demonstrate that the efficiency term contributes additional gains even when the agent relies solely on weight-based retrieval.

**Ablation Study on Different Graph-leveraging Methods** We conduct experiments to examine the impact of adaptive memory retrieval strategy of the H-EPM. Specifically, we consider two variants: (i) In the w/o adapt variant, the agent is required to recall episodic memory whenever a previously invoked tool connected by an edge contains relevant state information. If none of the connected edges provide state information, the agent defaults to weight-based retrieval. (ii) with GNN4Plan (Wu et al., 2024) traverses the H-EPM memory according to the original GNN4Plan strategy. Each variant is tested under identical conditions to isolate the effect of the ablated component. As shown in Table 4(C), both alternative graph-leveraging methods exhibit performance drops relative to the full H-EPM strategy. These results highlight the importance of dynamically invoking different memory types and incorporating state-level information.

in multi-turn agents: long-horizon decision-making under sparse feedback, where salient context is easily lost and effective tool-use patterns are difficult to transfer. By compressing successful trajectories into reusable episodic cues and procedural routines, H-EPM improves decision-making at inference and enables memory-assisted long-horizon exploration during RL.

## Impact Statement

This paper presents work whose goal is to advance the field of Machine Learning. There are many potential societal consequences of our work, none which we feel must be specifically highlighted here.

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

*Table 6.* Performance comparison across two evaluation metrics in agent multi-turn task ACEBench (process accuracy and end-to-end accuracy). Pro: Process. EtoE: end-to-end. RB: Reasoning Bank. RT: Retrieval Top-1.

| METRIC | MODEL | BASE | RB | TOOLNET | RT | H-EPM |
|---|---|---|---|---|---|---|
| PRO ACC | GPT-4.1-MINI | 0.633 | 0.701 | 0.627 | 0.641 | **0.719** |
| | GPT-4.1 | 0.653 | 0.703 | 0.664 | 0.601 | **0.765** |
| | GPT-4O | 0.664 | 0.684 | 0.659 | 0.628 | **0.720** |
| ETOE ACC | GPT-4.1-MINI | 0.400 | 0.483 | 0.436 | 0.473 | **0.533** |
| | GPT-4.1 | 0.503 | 0.498 | 0.481 | 0.432 | **0.567** |
| | GPT-4O | 0.484 | 0.477 | 0.473 | 0.467 | **0.503** |

# A. Related Work

## A.1. Graph-based Memory

Graph structures can organize memory and knowledge in a coherent, interconnected manner. They encode episodic or semantic information as nodes linked through relational edges, thereby enabling efficient retrieval of contextually relevant information (Liu et al., 2025). Graph-based methods continuously update the memory graph by adding new episodic events or semantic knowledge, dynamically creating nodes and relational links that refine the agent's understanding as it interacts with the environment (Xu et al., 2025; Anokhin et al., 2024; Jiang et al., 2025a). However, these approaches primarily focus on episodic memory. In tool-use tasks, tool dependencies can be determined by the inherent sequencing of tool invocations. In contrast to these graph-based methods, our approach encodes episodic state information on the edges and leverages these meaningful connections to jointly capture episodic relationships and tool dependency structures. The agent can rely on both episodic and procedural memory when making decisions.

## A.2. Tool Graph

Recent studies have explored tool graphs as a means of structuring and reasoning over large tool spaces. Depending on how the graph is constructed, existing methods can be categorized into four groups. First, approaches based on input–output relationships include ControlLLM (Liu et al., 2024b), Magnet (Yin et al., 2025), and NaviAgent (Jiang et al., 2025b), where nodes represent tools or resources and edges capture data dependencies. Second, methods leveraging historical trajectories, such as GTool (Chen et al., 2025b), ToolFlow (Wang et al., 2024), and NaviAgent, infer connections from past tool-use sequences, with ToolFlow further modeling parameter-level semantic similarities to generate coherent tool combinations. Third, manually constructed graphs exemplified by SciToolAgent (Ding et al., 2025) encode expert knowledge to guide LLMs in executing multi-step toolchains in scientific domains. Finally, approaches supporting dynamic updates, such as ToolNet (Liu et al., 2024a) and SGC (Wu et al., 2024), organize massive tool collections into weighted directed graphs and update them adaptively based on usage, enabling more efficient navigation and planning. By contrast, our method integrates state information with the tool graph during dynamic modeling, enabling the execution of tool graph following in a task-specific and stateful way.

# B. Case Study

Figure 3 illustrates the advantages of our method over existing memory based and tool graph based approaches in multi-turn, stateful tasks.

Figure 4 illustrates an example trajectory showing how H-EPM provides suggestions to guide agent decision-making.

# C. More Results

## C.1. Scenarios Without an Available Training Set

When a training set is absent, memory is updated online using test-time data following the procedure in Section 4.2, with successful trajectories identified in real time by an LLM-based judge. We evaluate this setting on the ACEBench multi-turn agent tasks, reporting end-to-end and process accuracy in Table 6. End-to-end accuracy is 1 if all target instance attributes are matched exactly, and 0 otherwise. Process accuracy measures alignment with the ideal function-call sequence, defined as

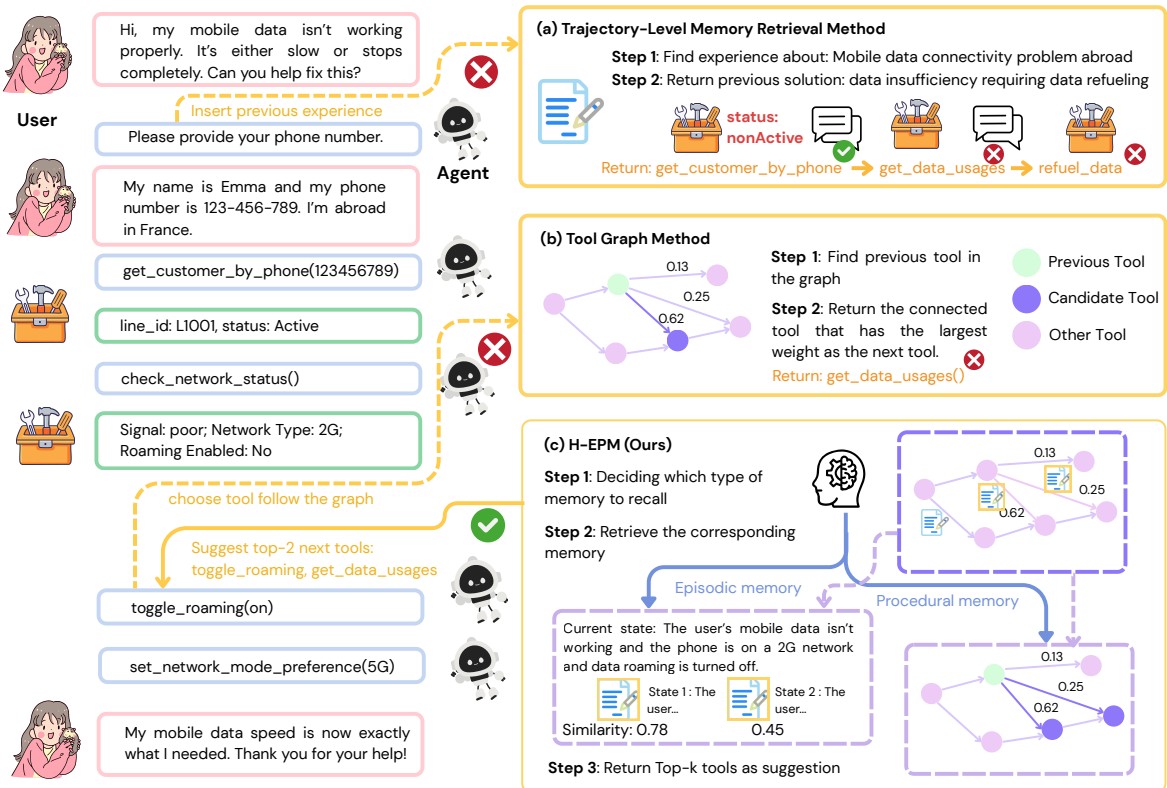

*Figure 3.* **Comparison with episodic memory and tool graph based methods in multi-turn tool-use tasks.** (a) In memory-based methods, memories are stored as discrete items, but these approaches typically incorporate only episodic event memories and retrieve trajectory-level experience before execution. Before the task objective becomes sufficiently clear, this retrieval process may return mismatched or inappropriate experiences, which can lead to task failure. (b) Tool graphs typically encode dependency relationships between agents, relying solely on tool dependencies extracted from previous experience. However, they lack consideration of task-specific states. (c) Our method represents both episodic and procedural memory within a unified structure and adaptively leverages these different types of memory at the state level. In this case, the actual issue lies in the mobile phone settings, and the agent can only confirm the underlying problem after invoking the information tool, which updates the state. However, (a) would retrieve a solution based solely on the observed symptoms before the agent has accurately identified the true cause, resulting in an incorrect response. (b), on the other hand, tends to select the tool that has been most frequently used in past trajectories or is highly connected in the tool-dependency graph. Because it ignores the current state and task-specific information, it also often leads to the selection of an incorrect tool.

$n/m$, where $m$ is the sequence length and $n$ is the number of correctly matched steps. ACEBench features multi-turn tasks with partially overlapping scenarios, where dynamic graph construction enables adaptive retrieval of state-specific memories or tool dependencies, yielding clear performance gains. In this setting, we replace Retrieve-Top-3 with Retrieve-Top-1, as retrieving multiple examples from a small task set often introduces weakly related trajectories that mislead decision-making. H-EPM remains effective with limited online memory, whereas ToolNet, which relies solely on tool dependencies, suffers a significant performance drop, highlighting that relevant prior experiences provide useful guidance while irrelevant memories introduce noise.

## C.2. Weight Parameter $c$

In H-EPM memory, the weighting function consists of accuracy and efficiency, where the efficiency term is controlled by parameter $c$. Table 7 reports the performance of GPT-4.1 on $\tau$-Bench. Parameter $c$ should strike a balance between efficiency and accuracy.

*Table 7.* Parameter $c$ in the weight.

| THE VALUE OF $c$ | 0 | 1 | 5 | 10 |
|---|---|---|---|---|
| $\tau$-BENCH | 0.660 | 0.710 | 0.695 | 0.678 |

*Table 8.* Token Entropy Statistics.

| POSITION | (I) SUMM TOOL | (II) BEFORE | (III) AFTER |
|---|---|---|---|
| ENTROPY | 0.305 | 0.659 | 0.140 |

### C.3. Token Entropy Analysis of the Summarization Tool

We analyze the behavior of the summarization tool by measuring token-level entropy across 30 trajectories. Specifically, we record (i) the first-token entropy of the summarization tool calling, (ii) the first-token entropy of the agent's response immediately before invoking the summarization tool, and (iii) the first-token entropy of the agent's response immediately after the summarization step. These values are illustrated in Table 8. The results show that token entropy decreases significantly after invoking the summarization tool, while entropy is substantially higher before its invocation.

### C.4. Implicit and Explicit Summarization Tool Invocation Decisions

We consider implicit and explicit strategies for controlling an agent's invocation of the summarization tool. (i) In the implicit strategy, the agent is prompted only at the beginning of inference and autonomously decides whether to invoke the summarization tool during the whole task. When a summarization invocation is detected, episodic memory is retrieved; otherwise, procedural memory is used. (ii) Explicit strategy directly prompts the agent at each turn to decide whether to perform summarization at the current step. and the selected decision determines which type of memory is recalled. As shown in Table 9, the explicit strategy achieves better performance but incurs higher computational cost (detailed in Appendix D). Under the explicit strategy, summarization tools are invoked more frequently, which leads to increased episodic memory retrieval and improved exploitation of state-specific information. For delay-sensitive scenarios, an implicit summarization tool invocation strategy can be adopted, yielding substantial performance improvements with an approximately 5% degradation compared to the explicit strategy.

### C.5. Foward Strategy in H-EPM Enhanced RL

In RL training, our method modifies the original policy by introducing a summarization tool and memory-based guidance. To ensure stable policy learning, the additional information introduced during rollout is removed during training. Specifically, we consider two strategies: (i) masking the extra information when computing the loss or (ii) directly discarding it prior to policy gradient computation. The results in Table 10 demonstrate that the masking strategy yields better performance.

Adding guidance tokens for exploration can introduce two distinct mismatches: 1) rollout–update mismatch, where the data are collected under a behavior policy that conditions on the extra tokens, but the policy is updated under a different conditioning signal; 2) train–test mismatch, where the model is optimized with access to guidance that will not be available at inference time. Two common treatments trade off these mismatches differently. Masking the guidance tokens in the loss keeps the parameter update aligned with the original (unguided) policy, largely eliminating the rollout–update mismatch, but it still leaves a train–test mismatch because the rollout trajectories were generated with additional information. In contrast, dropping the guidance tokens before computing policy gradients aligns training with inference-time inputs, but it creates a rollout–update mismatch because the behavior policy that produced the trajectories is no longer the one being optimized.

In our setup, guidance tokens are applied sparingly: over 70% of actions are executed without any added guidance. As a result, the train–test mismatch introduced by occasional guidance is limited in practice, whereas the rollout–update mismatch can be much more harmful if not controlled. This makes the masking strategy a better fit for our setting: it preserves a consistent update target while still benefiting from guided exploration. Furthermore, H-EPM's guidance is not arbitrary external advice. It is distilled from the agent's own previously successful trajectories, so it stays close to the policy's natural behavior distribution while nudging exploration toward higher-yield regions. This improves the success rate of rollouts and, in turn, leads to better final performance.

*Table 9.* Summarization tool invocation strategy.

| STRATEGY | $\tau^2$-BENCH | $\tau$-BENCH | TOOLSANDBOX |
|---|---|---|---|
| BASE | 0.358 | 0.658 | 0.633 |
| IMPLICIT | 0.484 | 0.685 | 0.679 |
| EXPLICIT | 0.512 | 0.710 | 0.704 |

*Table 10.* Foward strategy in H-EPM-E enhanced RL.

| SKIP RATE | $\tau^2$-BENCH | $\tau$-BENCH | TOOLSANDBOX |
|---|---|---|---|
| MASK | 0.223 | 0.544 | 0.510 |
| DISCARD | 0.219 | 0.489 | 0.494 |

## D. Computational cost

In the reinforcement learning setting, training a single epoch with our method requires approximately 20 hours on 8 H100 GPUs, which is comparable to the time cost of the GRPO baseline for the 4B model. When agent systems evaluated on the $\tau^2$-bench, stand inference takes about 50 minutes, while the runtime increases to about 60 minutes with our implicit summarization tool invocation strategy and to about 80 minutes with explicit summarization tool invocation strategy for the same 4B model. To construct the initial hybrid memory, we first perform three rollouts on the training set and retain the successful trajectory with the shortest number of turns. Using the Qwen3-4B model, processing approximately 2.2k tasks on the $\tau^2$-Bench requires about 15 hours. All inference experiments are conducted on one 80G A100 GPU.

## E. Implementation details

### E.1. Experiment details

**Details for Inference.**   When constructing the memory graph from the training set, we adopt an implicit summarization tool invocation strategy to reduce computational cost. When retrieving episodic memory, we compare the current state with previous state information retrieved from the edges of connected tools using the *all-MiniLM-L6-v2* embedding model and compute cosine similarity. After obtaining guidance from episodic memory, under the implicit strategy, the current state information is directly incorporated into the context to guide action selection. Under the explicit strategy, although the agent is prompted to decide whether to summarize the current state, the generated summary is used solely for comparison and is not included in the contextual input, as additional summarization may increase context length. In contrast, the implicit strategy invokes summarization less frequently, so incorporating the summary into the context does not significantly affect the context length. All inference results are obtained using the explicit strategy except those reported in the Table 9. Guidance from episodic and procedural memory is injected as a system prompt in the form of *"Suggested next tools: Tool_1, Tool_2"* when the top-2 tools satisfy the selection criteria, which is illustrated in the Appendix B.

**Details for RL.**   We specify several key parameters. The Kullback–Leibler (KL) divergence loss coefficient is set to $\beta = 0.001$. The training configuration uses one epoch, a batch size of 8, and 8 rollouts per task. Rewards are provided only upon successful task completion, and the advantage is computed solely from the outcome-based reward. We adopt the implicit summarization tool invocation strategy for H-EMP during training to reduce computational cost. All additional guidance and summarization content are excluded during training using two strategies (Appendix C.1), and all results for RL except those in Table 6 are obtained with the masking strategy.

### E.2. Summarization tool prompt

The summarization tool is used to help the agent summarize the current state. Rather than relying on a highly restrictive prompt, we allow the agent to autonomously determine the summarization content.

Prompt: *Please write a summary of the current state, including information from the environment and the user.*

### E.3. Summarization tool invocation implicit prompt

We allow the agent to autonomously decide when to invoke the summarization tool by adding the following instruction to the system prompt.

Prompt: *You must proactively summarize the current state and information of the task with the śummarize_the_taskíool and must call śummarize_the_taskát least once in every conversation. You should decide when to summarize the current state and information of the task with the śummarize_the_taskíool to help you make decisions about action. You **must** call it with a summary of the current state (including information from the environment and the user).*

### E.4. Summarization tool invocation explicit prompt

During inference, we try to increase the probability of invoking the summarization tool to allow H-EPM to provide more episodic memory.We explicitly prompt the agent to decide whether to invoke the summarization tool for the current state at each turn.

Prompt: *You are an intelligent assistant that needs to determine whether to call the śummarize_the_taskíool to summarize the conversation context.*

*Conversation context:* {*conversation context*}

*Please determine:Does the conversation context need to be summarized to better align with the current information for improved decision-making?*

*Please only answer "Yes" or "No", do not explain the reason.*

## F. Limitation

Our work provides guidance for agent decision-making in multi-turn tasks. However, models with stronger reasoning capabilities often exhibit weaker adherence to external guidance during execution, which limits the performance gains of our method compared to instruction-tuned models. Developing more effective mechanisms to align guidance with the reasoning processes of such models remains an important direction for future work.

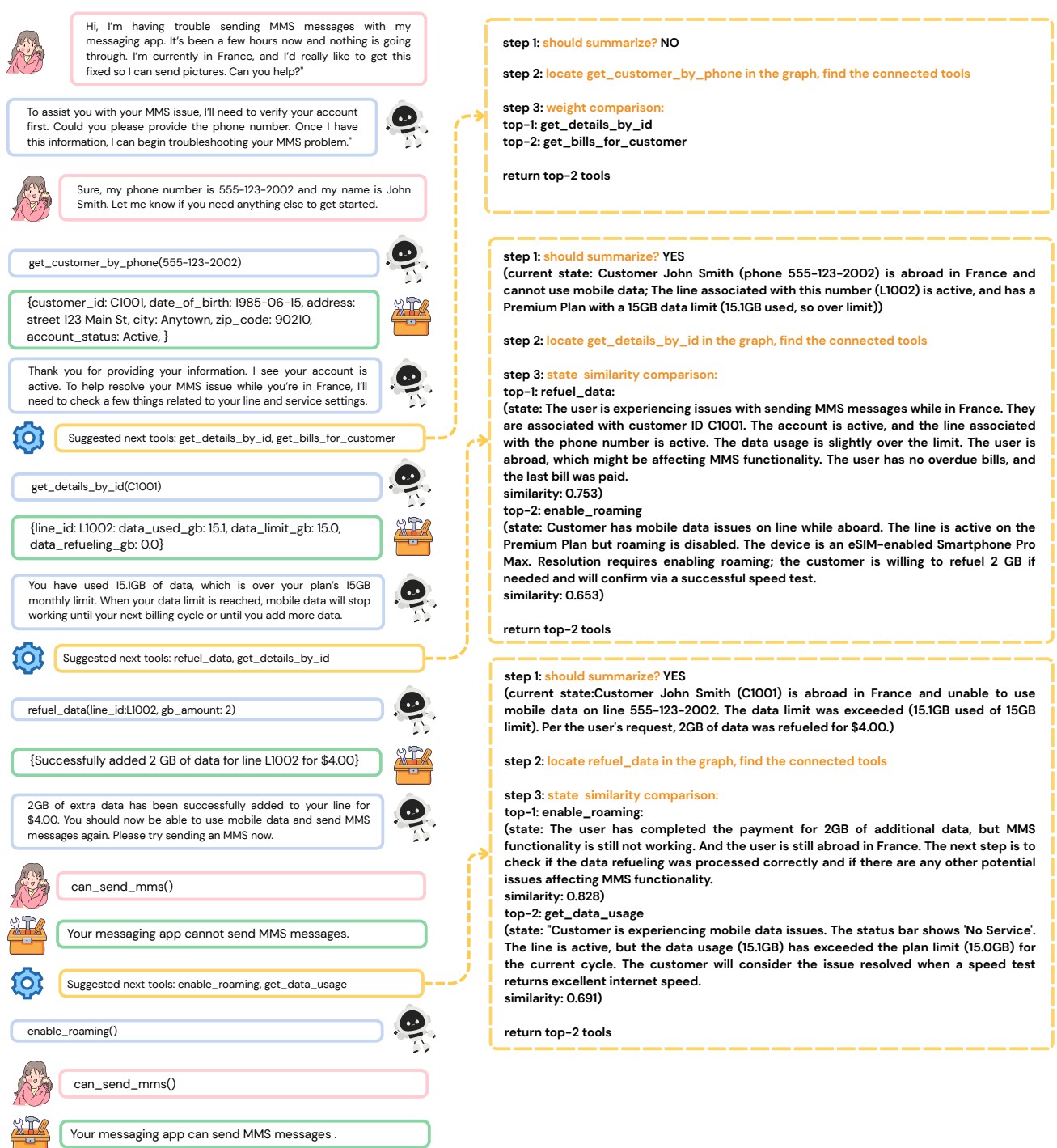

*Figure 4.* **An illustrative trajectory example produced by H-EPM.** In the trajectory, when the agent decides to invoke the summarization tool, it compares the current state with state information on connected edges to provide guidance via episodic memory. If the agent decides not to summarize the state at the current step, H-EPM provides guidance based on transition weights, corresponding to procedural memory.

