# OpenReview forum: "Experience-Evolving Multi-Turn Tool-Use Agent with Hybrid Episodic–Procedural Memory"
_ICML.cc/2026/Conference — ICML 2026 regular_

### Official Review · Reviewer_ZgCk · 2026-03-12

**Soundness:** 4
**Presentation:** 4
**Significance:** 4
**Originality:** 3
**Overall Recommendation:** 5
**Confidence:** 4

**Summary:**

This paper studies multi-turn tool-use agents under changing dialogue and environment state, and proposes H-EPM, a hybrid episodic-procedural memory mechanism. The method builds a tool-transition graph from successful trajectories, where edges store both procedural transition weights and compact state summaries, then uses this graph both at inference time for tool suggestion and during RL rollouts to bias exploration. Experiments on several multi-turn tool-use benchmarks show consistent gains over a base model and several memory/tool-graph baselines, and the paper also reports improvements when integrating H-EPM with GRPO-style RL training.

**Compliance With Llm Reviewing Policy:**

Affirmed.

**Final Justification:**

I am raising my score from 4 to 5. The paper presents a well-motivated and cohesive design that unifies episodic and procedural memory for both inference and RL in multi-turn tool-use agents, with solid empirical results across multiple benchmarks and model scales. My original reservations centered on memory graph dependence, initial bias, and path reinforcement. The rebuttal provided substantial new experiments (from-scratch memory construction, RL without pre-built graphs, memory coverage analysis at $50\%$/$75\%$/$100\%$, and GPT-5.1 baseline comparisons) that collectively addressed these concerns with concrete evidence. The only remaining issue, latency quantification, is a deployment-level detail that does not undermine the scientific contribution and can be addressed in revision.

**Key Questions For Authors:**

1. How does H-EPM handle tools that are not well covered by the memory graph, or tools that are entirely unseen during graph construction?
 The retrieval mechanism appears to assume that the current tool can be mapped to a graph node with meaningful outgoing candidates. It would be helpful if the authors could clarify the fallback behavior in low-coverage or unseen-tool cases.
2. How sensitive is the method to the quality and diversity of the initial graph construction?
 Since the graph is built from successful trajectories, I wonder whether early graph bias could affect later retrieval and RL exploration. Some discussion of how robust the method is to narrow or skewed initial memories would be valuable.
3. Does the online graph update process risk reinforcing early tool-transition preferences?
 Because successful transitions increase edge strength and stronger edges may in turn bias future rollouts, there may be a path-dependence effect. The skip-rate mechanism seems helpful, but I would appreciate more intuition or evidence on how the method avoids over-concentrating on a small set of familiar tool chains.
4. Can the authors comment more on the tradeoff between explicit summarization quality and runtime cost?
 The explicit summarization variant seems stronger, but also significantly more expensive. A brief discussion of when the additional cost is worthwhile would improve the practical takeaways of the paper.

**Limitations:**

No. The paper includes only a very brief impact statement, which does not meaningfully discuss limitations or potential negative societal impact. The authors could improve this by briefly discussing risks from biased or incomplete memory graphs, possible error reinforcement in tool suggestions, and privacy/security concerns when storing summarized interaction states.

**Strengths And Weaknesses:**

Strengths:
- The dual use of the memory graph for both inference-time guidance and RL exploration is a cohesive and well-motivated design for long-horizon tool-use agents.
- The paper introduces a simple but effective procedural weighting scheme that combines successful transition frequency with trajectory efficiency, which is a sensible way to reduce overuse of irrelevant tools.
- The proposed memory representation is more context-aware than pure tool-transition graphs, while remaining more structured and reusable than full trajectory retrieval.
- The empirical results are solid overall, with improvements across several benchmarks and model scales, and the RL integration is practically interesting.

Weaknesses:
1. Latency and computational overhead are nontrivial. As noted in the paper, the explicit summarization strategy introduces a noticeable runtime increase. While the accuracy gains are meaningful, the practical tradeoff between quality and efficiency could be discussed more carefully, especially for deployment settings where latency matters.
2. The method appears somewhat dependent on the quality and coverage of the constructed memory graph. Since the graph is built from successful trajectories, its usefulness may depend on how representative those trajectories are. The paper demonstrates good empirical performance, but a bit more discussion of robustness under sparse or biased initial memory would strengthen the presentation.
3. Some implementation and generalization details are left implicit. In particular, the paper could better clarify how the system behaves when graph coverage is limited, or when test-time tool usage patterns differ from those seen during graph construction. This does not invalidate the method, but clearer discussion would improve confidence in its applicability.

---

> ### Author Rebuttal · Authors · 2026-03-31
>
> We sincerely thank the reviewer for recognizing the effectiveness of our unified memory graph design for both inference and RL, as well as our strong empirical performance.
>
> ## W1&Q4 Latency and cost
> We thank the reviewer for highlighting this point. The additional runtime mainly comes from two components: (1) summarization tool invocations, which require extra forward passes to invoke and generate state abstractions. (2) memory retrieval and comparison, particularly the similarity computation between the current state and stored summaries. These steps introduce overhead compared to a standard inference pipeline without memory.
>
> We agree that latency is important for real-world deployment. This overhead can be mitigated through strategies such as adaptive summarization (triggered only under high uncertainty), lightweight or distilled summarization models, efficient retrieval, and top-$k$ pruning to limit comparisons. We will clarify this quality–efficiency tradeoff in the revision and note that our current implementation prioritizes effectiveness.
>
> ## W2 Initial memory
> We thank the reviewer for highlighting the point. We would like to clarify that this aspect has been partially evaluated in the paper. In particular, results on **ACEBench(Table 6 in the paper)** demonstrate that our method remains effective even when the memory graph is **initialized from scratch**, where **it is constructed online during testing from an empty graph and gradually updated with collected trajectories during test time**, suggesting a certain level of robustness.
>
> To further address this concern, we conduct additional experiments during the rebuttal on ToolSandbox, where the memory graph is **built entirely from scratch** using the original rollout trajectories. The results are shown below:
>
> |Model|Base|H-EPM|
> |-|-|-|
> |GPT-4.1|0.678|0.702|
> |Qwen3-4B|0.506|0.539|
>
> H-EPM still achieves consistent improvements **without a pre-constructed graph**, demonstrating robustness under limited or evolving memory. The gain is smaller than with a well-initialized graph (Table above), indicating that memory quality and coverage matter. As more trajectories are accumulated, performance improves progressively, showing the ability to self-refine over time.
>
> This aligns with our discussion in W3: memory is more effective when relevant past experiences are available. Accordingly, its benefits are limited under sparse coverage but increase as the memory expands. While H-EPM benefits from better initial coverage, it is not overly sensitive to it and can still adapt and improve in low-coverage settings.
>
> ## W3 & Q1 Generalization and limited coverage
> Please refer to our response to Reviewer kspL W1. H-EPM generalizes to unseen benchmarks through RL training. At inference, it uses lightweight memory that leverages abstracted states and tool patterns to reuse experience under partial overlap instead of exact matches.
>
> ## Q2  Initial graph
> For inference, please refer to our response to W2. For RL, we further conduct additional experiments where the memory graph is initialized from empty memory, rather than pre-constructed. The results are shown below:
>
> ||GRPO|Initial without graph|H-EPM|
> |-|-|-|-|
> |τ-Bench|0.510|0.532|**0.558**|
> |τ2-Bench|0.178|0.209|**0.213**|
>
> The results show that training from scratch is slightly weaker than using a well-initialized graph, indicating that initial memory quality matters. Nevertheless, H-EPM consistently outperforms the no-memory baseline on τ-Bench and τ²-Bench, demonstrating robustness under limited initial coverage.
>
> Overall, while strong initialization improves efficiency, H-EPM does not depend on it and can progressively build effective memory through RL as more trajectories are collected.
>
> ## Q3 Over-concentrating on a small set
> We thank the reviewer for highlighting this point. Our memory operates at a **state-aware level**, not just transition frequency. While edge weights capture procedural regularities, retrieval is conditioned on **state similarity**, allowing context-sensitive decisions rather than blindly following frequent transitions. Moreover, the summarization tool is used **sparingly during RL training** (≈1–2 times per \~18-step trajectory), thereby preventing over-reliance on a narrow set of behaviors and **maintaining a low guidance ratio to facilitate continued exploration**.
>
> In addition, the **skip-rate mechanism** introduces stochasticity by randomly skipping tool suggestions, preventing deterministic reliance on high-weight edges and reducing path dependence across rollouts. While our current design is effective, we agree that **memory pruning and diversification** could further improve robustness. Overall, **state-aware retrieval, sparse summarization, and stochastic exploration** together prevent collapse into narrow tool patterns while still leveraging past experience effectively.
>
> We hope these address your concerns and will include the valuable points in the revised manuscript.

---

> > ### Author Rebuttal · Reviewer_ZgCk · 2026-04-02
> >
> > I thank the authors for the thorough rebuttal. My main concerns (memory graph dependence W2, initial graph bias Q2, path reinforcement Q3) are well addressed by the new from-scratch experiments on both inference (ToolSandbox: GPT-4.1 from 0.678 to 0.702) and RL ($\tau$-Bench: from 0.510 to 0.532; $\tau^2$-Bench: from 0.178 to 0.209 without pre-constructed graph), the cross-benchmark generalization evidence (W3/Q1), and the state-aware retrieval combined with the skip-rate mechanism (Q3). The latency discussion (W1/Q4) remains qualitative but is a deployment-level concern that does not diminish the core contribution. I am raising my score from 4 to 5.

---

> > > ### Author Response · Authors · 2026-04-05
> > >
> > > We are glad to have addressed the reviewer’s main concern. We sincerely thank the reviewer for their time and valuable feedback. These discussions have helped improve the completeness and clarity of the paper, particularly in aspects such as initial graph construction, path reinforcement, latency, and cost.

---

### Official Review · Reviewer_Gkx7 · 2026-03-13

**Soundness:** 3
**Presentation:** 3
**Significance:** 2
**Originality:** 2
**Overall Recommendation:** 4
**Confidence:** 3

**Summary:**

The paper proposes H-EPM, a hybrid memory mechanism for multi-turn tool-use agents: procedural memory is a weighted directed tool-transition graph, and episodic memory is state summaries attached to edges and retrieved via embedding similarity. At inference, the agent gates between episodic retrieval (if matches pass selection criteria) and procedural execution (fallback to high-weight outgoing edges). H-EPM also biases RL rollouts toward historically successful transitions with a skip rate to preserve exploration. Experiments are reported on ToolSandbox, APIBank, and ACEBench.

**Compliance With Llm Reviewing Policy:**

Affirmed.

**Final Justification:**

The rebuttal addressed my main concerns.

**Key Questions For Authors:**

The method allows multiple state summaries to accumulate per edge. Can you clarify the number of summaries per edge, the number of tools, and the memory sizes? How does performance change as memory grows?
What are the exact selection criteria that decide episodic recall vs procedural execution?
For RL, the GPT5.1 base outperforms the older models significantly. How does the other method perform with GPT 5.1 (or newer/stronger model)?

**Limitations:**

Memory can keep growing (multiple summaries per edge), but there’s no clear stats on number of tools, edges, summaries/edge, total memory size, or how accuracy/runtime changes as it scales.
The episodic vs procedural switching rule is still a bit hand-wavy. what exact thresholds/criteria decide it?
RL results hinge on a strong model (GPT-5.1). I’d like to see how other baselines do on GPT-5.1 (or stronger) to tell if H-EPM still matters in the modern regime.

**Strengths And Weaknesses:**

Strengths: This is a logical hybrid between trajectory retrieval (coarse) and tool graphs (state-agnostic). It has interpretable memory structure and ablations support that both components matter.
Weaknesses:
- The paper does not thoroughly characterize memory growth (summaries per edge, pruning/eviction), retrieval latency as memory scales, or stability under distribution shifts
- Summaries incorporate information from user + environment/tool outputs, which can include adversarial instructions.
- RL comparisons do not isolate whether the gains come specifically from the hybrid episodic–procedural structure versus simpler heuristics

---

> ### Author Rebuttal · Authors · 2026-03-31
>
> We sincerely thank the reviewer for recognizing our methodological contributions, particularly the design of H-EPM and the thoroughness of the experimental evaluation.
>
> ## W1
> We thank the reviewer for raising the following points.
>
> **Memory growth**
>
> In our framework, memory is organized as a **structured tool graph**, which naturally limits its growth. Even with \~20k trajectories on $\tau^2$-Bench, the memory remains compact (32 nodes, 133 edges, 1124 summaries), as it relies on **edge-level aggregation** rather than growing linearly with data. We agree that memory can be further compact by methods such as **merging similar summaries and retaining only representative states**, allowing us to maintain a bounded and informative memory without sacrificing performance.
>
> **Retrieval latency**
>
> Retrieval in H-EPM is efficient as it is localized to the adjacent edges of the current tool node, rather than scanning the entire memory. The complexity is thus governed by the ​**local branching factor**​, which is small in practice. Even with state-summary comparisons, operations remain confined to neighboring edges, ensuring low latency and scalability as memory grows. **Further details are provided in Appendix D, and additional discussion can be found in our response to Reviewer ZgCk W1.**
>
> **Stability under distribution shifts**
>
> Please refer to our response to Reviewer kspL W1 on generalization and to Reviewer ZgCk Q2 on initialization from an empty graph. These results **indicate that our method exhibits stable behavior under OOD generalization and dynamic graph updates.**
>
> ## W2 Summarization tool
>
> **We clarify that the summarization tool is intended to mitigate, rather than amplify, adversarial or irrelevant information.** It compresses raw, potentially noisy interactions into ​**abstracted, task-relevant state representations**​, filtering out spurious details. These summaries are then used in a ​**retrieval-based manner**​, where decisions are guided by consistency with past successful experiences. ​**This provides an additional layer of robustness**​, as the impact of any single noisy or adversarial interaction can be moderated through comparison with ​**historically validated states**​. Overall, the tool acts as a **denoising and abstraction mechanism** that improves robustness to noisy inputs.
>
> ## W3 Simple heuristics
>
> We compare with **AgentEvolver (Table 3 in the paper),** which can be viewed as a trajectory-level reuse heuristic without structured decomposition. In addition, we include an ablation using **only the summarization tool during RL (in response to Reviewer kspL Q3)**, which can also be regarded as a simple heuristic. Empirically, our full method consistently outperforms both, suggesting that the gains are more likely from the H-EPM design than to simple heuristics.
>
> ## Q1 The memory sizes
>
> Please see answer to W1.
>
> ## Q2 The performance as memory grows
>
> We thank the reviewer for this valuable suggestion. To further investigate the impact of memory coverage, we conduct additional experiments on τ2-Bench by varying the proportion of training data used to construct the memory (50%, 75%, and 100% with H-EPM). The results are:
>
> |Model|Base|50%|75%|100%|
> |-|-|-|-|-|
> |GPT-4.1 |0.358|0.487|0.508|**0.512**|
> |Qwen3-4B|0.158|0.219|0.228|**0.232**|
>
> H-EPM achieves **clear improvements over the base model even with only 50% of the data**, demonstrating robustness under sparse memory coverage. Performance increases steadily as more data is added, indicating that **memory diversity and coverage are important** for maximizing effectiveness. H-EPM benefits from richer experience while remaining effective in low-resource settings.
>
> We will include this analysis in the revised manuscript to better clarify the robustness of our approach.
>
> ## Q3 The selection criteria of memory type
>
> We clarify that the selection mechanism is defined in Section 4.3: the agent first decides whether to invoke the summarization tool. If invoked, the resulting summary is matched against stored summaries (episodic retrieval); otherwise, the agent relies on edge weights (procedural selection).
>
> Invoking summarization signals a need for clearer state understanding, making state-based retrieval from similar experiences beneficial. Please see more analysis in our answer to Reviewer kspL Q2.
>
> ## Q4 For RL, the GPT5.1 outperforms the older models.
>
> We clarify that **GPT-5.1 is not used in any part of the RL training process**. Both memory construction and rollout generation are conducted using the same backbone model being trained(Qwen3-4B-Instruct-2507), ensuring that all reported improvements come from our method rather than stronger external models. GPT-5.1 is included **only for inference-time evaluation**, to show that H-EPM can further improve even strong models.
>
> ## Q5  Results with GPT 5.1
>
> Please see answer to Reviewer kspL W2.
>
> We hope these address your concerns and we will include the valuable points in the revised manuscript.

---

> > ### Author Rebuttal · Reviewer_Gkx7 · 2026-04-04
> >
> > My concerns have been adequately addressed, and I am adjusting my score to 4.

---

> > > ### Author Response · Authors · 2026-04-05
> > >
> > > We are glad to have addressed the concerns raised by the reviewer. We sincerely thank the reviewer for their valuable suggestions (e.g., regarding memory growth and stability under distribution shifts), which we believe have helped improve this work.

---

### Official Review · Reviewer_kspL · 2026-03-13

**Soundness:** 3
**Presentation:** 3
**Significance:** 3
**Originality:** 3
**Overall Recommendation:** 4
**Confidence:** 4

**Summary:**

The paper proposes H-EPM, a hybrid episodic-procedural memory framework for multi-turn tool-use agents. It constructs a state-annotated tool-transition graph from historical trajectories, capturing both procedural tool dependencies (edge weights) and episodic contexts (state summaries). The agent adaptively utilizes an autonomous summarization tool to switch between episodic recall and procedural execution. The method is evaluated in both inference and RL training settings across multiple benchmarks.

**Compliance With Llm Reviewing Policy:**

Affirmed.

**Key Questions For Authors:**

1. Does introducing the explicit summarization tool affect the model's normal task-solving strategy? In Appendix C.3, the token entropy drop is attributed to the summarization action. However, could this drop simply be an artifact of injecting the retrieved Top-k candidate tools into the context, which artificially increases the certainty of the next action?
2. Intuitively, context summarization and episodic retrieval are two independent intents. By coupling them together, can the model genuinely learn the strategy of calling the summarization tool specifically to retrieve similar episodes? Does this tight coupling introduce limitations or rigidness in the agent's instruction-following behavior?
3. The skip operation during RL training creates a distribution gap in the expected returns of the summarization action between training (where procedural guidance is randomly skipped) and testing (where it is not). Please analyze how this train-test gap affects the model's policy learning and whether it biases the agent to over-rely on the summarization tool during training.

**Limitations:**

Yes

**Strengths And Weaknesses:**

## Strengths
1. The paper addresses a critical challenge in multi-turn tool use: effectively reusing past experiences in long-horizon contexts without overfitting to full, rigid trajectories.
2. The integration of episodic and procedural memory into a unified graph structure is logically designed. The idea of using an autonomous state-summarization tool to dynamically trigger episodic retrieval is innovative and elegant.

## Weaknesses
1. The proposed memory system heavily relies on the accumulation of highly similar experiences (e.g., the specific scenario in Fig 4). Because the graph uses specific tools as nodes, it is inherently restricted to previously seen toolsets (In-Distribution). This explains the reliance on building the memory graph directly from the training set. The paper lacks a thorough discussion of this limitation and fails to provide comparative analyses with baselines regarding cross-domain generalization or adaptation to unseen toolsets.
2. As the authors noted, there are few experiments based on recent LLMs with strong native reasoning capabilities. The significant performance gains demonstrated primarily on models like GPT-4o and GPT-4.1-mini are less convincing for state-of-the-art reasoning models, which might not require such explicit external memory guidance.

---

> ### Author Rebuttal · Authors · 2026-03-31
>
> We sincerely thank the reviewer for recognizing our work on a critical challenge in multi-turn tool use and the clarity and elegance of our method.
>
> ## W1 Generalization
> First, we provide empirical evidence of generalization beyond the training distribution. Although RL training uses **only \$\\tau\$-Bench**, consistent improvements on **ToolSandbox and \$\\tau^2\$-Bench** show that the learned policy and memory usage strategy transfer to unseen tools and OOD settings, suggesting H-EPM captures **generalizable decision patterns**.
>
> At test time, memory is lightweight and experience-driven, and it is effective only when there is **meaningful overlap between past and current situations**—similar to how humans cannot use swimming experience to guide car repair. Our method leverages memory under this principle, and to further improve utilization, we use abstracted state summaries and tool-transition patterns, enabling reuse across **partially similar scenarios** rather than requiring exact matches.
>
> ## W2 Strong models
> Table 2 in the paper included results with **GPT-5.1 high reasoning mode**, a model with strong reasoning capabilities. H-EPM continues to provide improvements, indicating that even capable models benefit from leveraging past experience. During rebuttal, we also include **GPT-5.1-based baselines** (ReasoningBank and ToolNet) for a fairer comparison:
>
> |Method|τ2-Bench|τ-Bench|ToolSandbox|
> |-|-|-|-|
> |GPT-5.1(Base)|0.842|0.739 |0.647|
> |GPT-5.1+ReasoningBank|0.872|0.681|0.634|
> |GPT-5.1+ToolNet|0.894|0.752 |0.661|
> |GPT-5.1+H-EPM (ours)|**0.921**|**0.791**|**0.670**|
>
> **H-EPM consistently outperforms all baselines** across benchmarks. Notably, the consistent gains over a strong base model suggest that complementing reasoning with experience reuse can **provide additional benefits**. Compared to other memory- or graph-based approaches, our method achieves more effective integration of past experience, leading to better decision-making.
>
> ## Q1 Entropy and certainty
> First, We clarify that Appendix C.3 **does not treat entropy reduction as an objective, but uses entropy as a diagnostic signal to analyze when and why the agent invokes the summarization tool**. The agent tends to call the tool under high entropy, indicating a need to extract and structure state information for better decisions.
>
> **Our goal is not to reduce entropy or increase confidence, but to improve decision quality and overall task success rate.** Injecting Top-$k$ candidate tools provides structured guidance, and all improvements are evaluated by end performance rather than entropy. Tables 1 and 4 in the paper show that both the summarization tool and Top-$k$ suggestions improve success rates, enabling better strategy to make decisions.
>
> ## Q2  Summarization with episodic retrieval
> First, we clarify that our method is **not a simple combination of summarization and episodic retrieval**, but a **tightly integrated design where summarization enables effective episodic memory**. Adding summarization to other baselines does not yield comparable gains (Response to Reviewer Pzad Q1), indicating that it is not a standalone component but provides structured state representations crucial for retrieving relevant experiences.
>
> Second, our design is motivated by the following intuition: **when the agent decides to invoke the summarization tool, it indicates that the current decision requires a more explicit understanding of the underlying state**. In such cases, it is natural to **leverage memory from similar successful past states to guide decisions**. Thus, coupling summarization with episodic retrieval aligns with its role as a trigger for state-aware reasoning, where retrieving relevant experiences is particularly beneficial.
>
> To avoid over-reliance, we provide **Top-$k$** candidate tools as suggestions rather than enforcing actions, preserving **flexibility** for context-dependent decision-making.
>
> ## Q3  Skip operation
> We clarify that H-EPM (including the skip mechanism) is used only during RL training; **at test time, we evaluate the learned policy without any memory or skip-based guidance**. This design shows that H-EPM improves the **policy itself rather than relying on external memory at inference**, enabling OOD generalization even in scenarios where no memory can be constructed, as supported by the empirical results.
>
> During training, memory guidance is applied sparsely (over 70% of actions proceed without it), mitigating potential bias introduced by the train-test difference.
>
> ||GRPO|only summary|H-EPM|
> |-|-|-|-|
> |τ-BENCH|0.510|0.524|0.558|
> |τ2-BENCH|0.178|0.189|0.213|
>
> To examine potential over-reliance on summarization, we conduct additional experiments. Summarization alone gives modest gains over GRPO, refining information but not fully leveraging past experience. H-EPM achieves larger improvements by integrating summarization with structured memory retrieval.
>
> We hope these address your concerns and are glad to discuss further.

---

> > ### Author Rebuttal · Reviewer_kspL · 2026-04-03
> >
> > Thanks for the detailed response and it addresses most of my concerns. The setting of applying H-EPM only in RL training phase and the results showing that it improves the policy are interesting. Yet the generalizability of H-EPM in RL training does not change the limitation of it as a memory system (for inference) relying on similar experiences of seen toolsets.

---

> > > ### Author Response · Authors · 2026-04-05
> > >
> > > We thank the reviewer for raising this important point regarding generalizability at inference time. We respond from the following aspects:
> > >
> > > ---
> > >
> > > ### “Initial empty graph” construction
> > >
> > > We further validate this property by constructing the memory graph **from scratch (empty initialization)**.
> > >
> > > In this setting the agent progressively builds the graph through test-time interaction and no pre-existing tool structure is provided.
> > >
> > > Please refer to the detailed results on **ACEBench (Table 6 in the paper)** and our response to Reviewer ZgCk W2&Q2. The performance improvement under this setup shows that H-EPM is **not dependent on a fixed or pre-covered toolset** and can **adaptively incorporate new tools through experience accumulation**. Therefore, it naturally supports **the addition of new tools beyond those initially seen**.
> > >
> > > ---
> > >
> > > ### Generalization via tool description (new experiment)
> > >
> > > To directly address the reviewer’s concern, we conduct an additional experiment where *tool identity is no longer tied to exact names*.
> > >
> > > Specifically, we build the graph on **τ²-bench (Telecom)** tasks and then transfer the learned memory to **ToolSandbox phone-setting tasks** (164 tasks with unseen tools).
> > >
> > > **Experimental Setting**
> > >
> > > **Summarization:** Only a summarization tool is provided, without constructing a memory graph.
> > >
> > > **Original Graph:** The graph is constructed using ToolSandbox trajectories and updated during execution.
> > >
> > > **τ² Graph (without update):** The graph is constructed from τ²-bench. During inference, tools are retrieved via description-based matching, while the graph structure remains fixed (no updates).
> > >
> > > **τ² Graph (with update):** Tool names are replaced with **tool descriptions** as graph nodes. Tools are matched based on description similarity (threshold 0.4). If a new tool has similarity below 0.4 with all existing nodes, a new node is created, enabling dynamic expansion of the graph.
> > >
> > > **Results:**
> > >
> > > | Model| Base  | Summarization | Original Graph | τ² Graph (w/o update) | τ² Graph (with update) |
> > > |--|-|--|--|--|-|
> > > | GPT-4.1    | 0.613 | 0.646 | 0.710 | 0.693 | 0.698 |
> > > | Qwen3-4B   | 0.491 | 0.515 | 0.573  | 0.551  | 0.556 |
> > >
> > > These results show that while summarization alone yields modest improvements over the base model, incorporating memory leads to substantially greater gains. Even without graph updates, description-level matching enables effective transfer across toolsets, consistently outperforming the base model when the graph is constructed from a different benchmark. Allowing dynamic graph expansion further improves performance, demonstrating that H-EPM can flexibly incorporate unseen tools.
> > >
> > > Overall, the experiment indicates that H-EPM generalizes to unseen tools through **semantic abstraction rather than exact tool matching**, while maintaining strong performance under cross-benchmark transfer.
> > >
> > > ---
> > >
> > > ### Existing tool-graph methods
> > >
> > > We agree that memory-based methods may rely on seen tool patterns. However, this limitation is largely unaddressed in prior tool-graph approaches. Existing methods typically: (i) assume a fixed tool set and do not explicitly study generalization to unseen tools[1,2], or (ii) require additional training[3] or re-collection of trajectories when new tools are introduced[4,5].
> > >
> > > Our method mitigates this limitation in the following ways. H-EPM does not assume a static tool graph; instead, it supports incremental graph construction and updates, allowing new tools to be integrated without retraining the policy. Moreover, our RL formulation enables the learned policy to generalize to new tools after training. In addition, the tool-description-based generalization experiment further demonstrates the robustness of our approach to unseen tools, particularly when they share partially overlapping or semantically similar experiences.
> > >
> > > [1]Can Graph Learning Improve Planning in LLM-based Agents.
> > > [2]Graph RAG-Tool Fusion.
> > > [3]GTool: Graph Enhanced Tool Planning with Large Language Model.
> > > [4]ToolNet: Connecting Large Language Models with Massive Tools via Tool Graph.
> > > [5]ControlLLM: Augment Language Models with Tools by Searching on Graphs.
> > >
> > > ### Summary
> > >
> > > While memory systems inherently benefit from relevant past experience, our method supports **dynamic integration of new tools without retraining**. Experiments results demonstrate robustness under **empty-graph construction**, and enables **cross-tool generalization via description-level matching**, with consistent empirical gains.
> > >
> > > Overall, H-EPM is **not restricted to exact tool reuse**, but can generalize to **partially overlapping experiences and unseen toolsets**. We hope this helps address the reviewer’s concern.

---

### Official Review · Reviewer_Pzad · 2026-03-14

**Soundness:** 4
**Presentation:** 4
**Significance:** 3
**Originality:** 4
**Overall Recommendation:** 5
**Confidence:** 4

**Summary:**

This paper introduces a hybrid episodic–procedural memory strategy (H-EPM) that enables experience-induced self-evolution of multi-turn tool-use policies, by adaptively reusing partially overlapping successful experiences in both inference and training. The emprical evaluation proves the strategy is effective.

**Compliance With Llm Reviewing Policy:**

Affirmed.

**Key Questions For Authors:**

Please refer to my questions and concerns above.

**Strengths And Weaknesses:**

Strengths:
1. This paper targets an important problem in multi-turn tool-use agents, i.e., facing partial observability, evolving environment state, and long-horizon sparse-feedback issues. Meanwhile, the authors have done a great job in the motivation explanation and method introduction.
2. The method is relatively simple and implementable. Although the novelty looks more integrative than fundamental, the method design is still impressive, i.e., the graph construction, edge weights, episodic annotations, and adaptive retrieval logic are easy to understand and not overloaded with unnecessary machinery.
3. I think one of the most impressive contribution in this work is to use the same memory idea for both inference and RL, which is a good design choice. A lot of papers stop at test-time retrieval tricks. Here, they also use memory to bias exploration during RL, which makes the work feel more complete.

Weakness:

I do think this paper has done a great job. However, I have some concerns/questions for the authors:
1. First, I am not sure about **the real effectiveness of the “hybrid memory”.** I doubt that the summarization tool itself already helps a lot. In Table 4, “w/o memory” still improves substantially over the base model, especially on $\tau^2$-Bench (0.358 → 0.430). That suggests a nontrivial fraction of the gain may come from adding a summarization mechanism and extra structured prompting, not from the graph memory. More importantly, this creates **a fairness question for the baselines.** If H-EPM gets an autonomous summarization tool plus memory-guided candidate restriction, while baselines mainly get retrieval or graph traversal, then some of the comparison may not be apples-to-apples.
2. The method may be overstating “self-evolution.” What I really see is an agent that stores successful trajectories, compresses them, and then uses them to bias future action selection and RL rollouts. That is useful, but it is closer to memory-augmented policy guidance than to a deeper notion of autonomous self-evolution.
3. Based on point 2, **the memory only seems to learn from successes**, which can bias it toward frequent or easy pathways. The graph is built from successful trajectories, and the weights are based on successful transitions. That ignores informative failures, near-misses, and dead ends, which are often exactly what a long-horizon agent needs to avoid repeating.
4. The paper shows aggregate improvements, but **multi-turn memory systems can fail** through stale summaries, wrong recall, conflict between episodic and procedural cues, and compounding early mistakes. I wanted explicit failure cases and negative examples. The current ablations are useful but not enough on this point.
5. In experiment section, I notice that the method requires dataset-specific tuning of p_skip. More specifically, ToolSandbox prefers 0.8, $τ-$Bench prefers 0.9, and $\tau^2-$Bench prefers 1.0. That tells me the method is not fully plug-and-play; it depends on knowing the degree of procedural dependence in the domain.
6. On the other hand, The ToolSandbox split is created by the authors. Since ToolSandbox only has a test set, they split it into roughly 800/200 to construct memory. That is understandable, but it means a chunk of the empirical evidence depends on a custom split rather than a standard benchmark protocol.

---

> ### Author Rebuttal · Authors · 2026-03-31
>
> We sincerely thank the reviewer for recognizing the significance of the problem setting, our motivation, methodological design, and the unified use of memory for both inference and RL.
>
> ## Q1 The summarization tool
> We agree that the summarization tool alone brings improvements. However, it is essential for constructing meaningful state abstractions for episodic memory. More importantly, H-EPM provides substantial gains beyond summarization. Table 4(A) shows that H-EPM adds up to 19.7% gain over the summarization tool alone, highlighting the benefits of hybrid memory compared with using the summarization tool alone.
> To ensure fairness, we also equip baselines with the summarization tool:
>
> |Method|τ-Bench|τ²-Bench|ToolSandbox|
> |-|-|-|-|
> |only + sum|0.667|0.430|0.647|
> |ToolNet|0.652|0.425|0.643|
> |ToolNet + sum|0.670|0.441|0.655|
> |ReasoningBank|0.645|0.413|0.605|
> |ReasoningBank + sum|0.632|0.408|0.612|
> |H-EPM|0.710|0.512|0.704|
>
> These results show that **simply adding a summarization tool yields limited or inconsistent gains** and can even degrade performance. A key reason is that prior methods (e.g., ReasoningBank) rely on stored experiences that are **not well aligned with online summaries**, introducing inconsistency and noise. In contrast, H-EPM integrates summarization into both memory construction and online state abstraction, ensuring consistent representations and more stable performance gains.
>
> ## Q2 Self-evolution
> We thank the reviewer for this clarification, and we would like to emphasize that our notion of self-evolution is specifically grounded in experience-induced self-evolution, where the system evolves through the accumulation and refinement of experience memory, rather than a fully autonomous form of self-modification.
>
> Second, within the RL framework, **memory and policy are updated in a coupled manner**: the agent collects new trajectories, incorporates successful experiences into memory, and leverages the updated memory to guide subsequent exploration and optimization. This **closed feedback loop between experience accumulation and policy updating** is what we term “experience-induced self-evolution.” We acknowledge that **“experience evolution”** may be a more precise term, and we will clarify this in the revised version.
>
> ## Q3 Fail experience
> Thank you for the insightful suggestion. We agree that incorporating failure-aware signals is a promising and important direction.  Our design focuses on learning from successful trajectories, which provide clear supervision and allow the agent to reuse successful **effective procedural patterns** and **contextually relevant decisions**. Leveraging failure trajectories is **nontrivial**, as failures can arise from diverse causes (e.g., missing information or wrong tool choice), making it hard to extract actionable guidance. Effectively using such signals would require more sophisticated mechanisms, like failure attribution or counterfactual reasoning, which we leave as important future work.
>
> ## Q4 Failure case
> A failure occurs when the current task is absent from memory, causing the agent to rely on superficially similar cases. Despite the apparent similarity, the correct actions may differ. For example, in retail, a user may request an exchange for an ineligible item that should be returned, yet the agent may still suggest an exchange based on similar memory. Mitigation strategies include uncertainty-aware mechanisms to defer or clarify decisions and constraint-aware validation before action selection.
>
> ## Q5  p\_skip
> The analysis over different $p_{\text{skip}}$ values highlights robustness rather than sensitivity: while knowing the degree of procedural dependence in a domain can further help select $p_{\text{skip}}$​ for optimal performance, we emphasize that such tuning is ​**not required**​, since H-EPM consistently improves performance across a broad range $p_{\text{skip}} \in [0.8,1.0]$, demonstrating its robustness as a plug-and-play component. In RL, the policy is trained only on $\tau$-Bench, while gains on ToolSandbox and $\tau^2$-Bench come purely from generalization. As shown in Table 3 in the paper, H-EPM consistently improves performance on unseen benchmarks, indicating a transferable exploration strategy.
>
> ## Q6 ToolSandbox split
> We thank the reviewer for raising this important point. The split is **randomly constructed**, and **all methods are evaluated on the same tasks**, ensuring a **fair and controlled comparison**. Its purpose is solely to enable **memory construction** in the absence of an official training set and does **not favor H-EPM**. To further validate this, we evaluate RL-based methods on the **full ToolSandbox dataset without splitting**. As shown below, H-EPM-enhanced RL still outperforms GRPO, confirming that our conclusions **do not rely on the custom split**.
> ||base|GRPO|H-EPM(p_skip=0.8)|
> |-|-|-|-|
> |ToolSandbox|0.509|0.519|0.535|
>
> We hope these address your concerns and are glad to discuss further.

---

> > ### Author Rebuttal · Reviewer_Pzad · 2026-04-06
> >
> > Thanks for the detailed response and it addresses most of my concerns. I believe this is a solid work, and I will maintain my score.

---

> > > ### Author Response · Authors · 2026-04-06
> > >
> > > We are glad that our responses have addressed the concerns. We thank the reviewer for their time and the helpful suggestions(e.g., the summarization tool, fail experience and failure case), which have improved the quality of our work.

---

### Decision · Program_Chairs · 2026-04-30

**Decision:**

Accept (regular)

**Comment:**

Two modes of failure of tool-use agents performing complex multi-turn tasks are: a) over reliance on episodic memories (trajectory retrieval, which might be too specific to a different task and not generalize well to the current task); and b) over reliance on procedural memories (a graph describing typical sequences of tool use, which does not account for changing environmental state or explain why a tool was used). This work tries to bridge this gap with the hybrid episodic-procedural memory (H-EPM), which combines the strengths of both. Beyond this being helpful during inference, H-EPM can help reinforcement learning (RL) by biasing exploration in long trajectories towards higher reward regions of the space. Substantial gains in multi-turn tool-use benchmarks.

This work targets some of the important problems in current multi-turn tool-use agents: Effectively reusing past experiences in a long-horizon context without overfitting, and addressing sparse rewards in long trajectories, evolving environment states, and partial observability. The design of the hybrid structure between episodic and procedural memory is well-motivated and successfully captures advantages of both. The same hybrid memory idea is used for both inference and RL, and it produces benefits in both cases, unlike typical test-time scaling methods that require ad-hoc procedures. This makes the work more cohesive and shows the generality of the core approach.

The empirical evaluation is solid and shows significant gains across the board (over a variety of benchmarks and models). Given the increasingly widespread use of multi-turn agents, this contribution can have a significant practical impact. Overall, a paper that merits acceptance.